# Short-Term Effects of Dry Needling with a Standard Exercise Program on Pain and Quality of Life in Patients with Chronic Mechanical Neck Pain

**DOI:** 10.3390/jcm11206167

**Published:** 2022-10-19

**Authors:** Muhannad Almushahhim, Shibili Nuhmani, Royes Joseph, Wafa Hashem Al Muslem, Turki Abualait

**Affiliations:** 1Department of Physical Therapy, College of Applied Medical Sciences, Imam Abdulrahman Bin Faisal University, P.O. Box 2435, Dammam 31451, Saudi Arabia; 2Department of Pharmacy Practice, College of Clinical Pharmacy, Imam Abdulrahman Bin Faisal University, Dammam 34212, Saudi Arabia

**Keywords:** neck discomfort, injection, musculoskeletal condition, neck pain

## Abstract

Background: This study aimed to determine the short-term effects of dry needling (DN) combined with a standard exercise program on pain and quality of life in patients with chronic mechanical neck pain (CMNP). Methods: Thirty-one patients with CMNP were randomly allocated to the experimental and control groups. The experimental group received DN and underwent a standard exercise program (one DN session and six exercise sessions) for two weeks, whereas the control group underwent the same exercise program alone for two weeks. The participants’ scores in the Numeric Pain Rating Scale (NPRS), Neck Disability Index (NDI), Short Form-36 Quality of Life Scale (SF-36 QOLS), and Beck Depression Inventory (BDI) before and after the intervention were assessed. Results: The control and experimental groups’ post-test NDI, NPRS, SF-36 QOLS, and BDI scores significantly differed from their baseline scores (*p* ≤ 0.05). The between-group comparison of the post-test scores using Wilcoxon rank-sum test revealed no significant differences between the NDI, NPRS, BDI, and SF-36 QOLS scores of both groups (*p* ≥ 0.05). Conclusions: One session of trigger point DN (TrP-DN) with exercise and exercise alone showed the same pain and quality-of-life outcomes after a two-week intervention. We did not recognise TrP-DN as an efficient intervention, not because we obtained evidence that it is ineffective, but because there were inadequate high-quality studies on the subject and unavailable data on the minimum quantity of injections required for better DN outcomes in CMNP patients.

## 1. Introduction

Neck pain is one of the most common musculoskeletal conditions in the general population, with a prevalence rate ranging from 16.7 to 75.1% and a lifetime prevalence of 49% [1]. Chronic mechanical neck pain (CMNP) is a type of neck and/or shoulder pain with mechanical features, such as symptoms that are made worse by cervical motions and by maintaining a forward-facing posture for extended periods. Cervical pain with or without radiation that has not been recognised by pathological aetiology is known as CMNP and as non-specific neck pain. The upper trapezius, elevator scapulae, splenius, and multifidus are some of the most frequently impacted muscles [2,3]. At least 64% of people in the population, according to Côté, Cassidy [4], experience neck pain. CMNP affects 20% of the adult population on a mean prevalence weighted level and increases with age (62% of adults >75 years and 32% of adults between 25 and 34 years). This is greater than the proportion of people with diabetes or asthma in the same demographic [5]. Moreover, CMNP related to disability and long-term follow-up are becoming more common. About 50% of those with neck discomfort have persistent symptoms and moderate disability, which results in significant financial costs [6] and may lead to poor quality of life. Geneen, Moore [5] et al. and Leadley, Armstrong [7] found that CMNP patients have a lower quality of life and that lifestyles influenced the prevalence rates of cervical disorders. A higher incidence of anxiety and depression symptoms was reported in patients with neck pain [8]. Studies by Chen, Wang [9] and Zhang, Cheng [10] reported a significant difference in anxiety and depression scores between healthy and neck pain patients revealing a close association between neck pain depression and anxiety symptoms. However, the specific pathology of CMNP is still unclear because there are a small number of known cases of CMNP, even of trauma or severe degenerative conditions. Neck pain is thought to have a wide range of causes [11]. Patients with CMNP frequently experience hyperalgesia in their muscles, skin, and ligaments during dynamic and passive motions, as well as when being palpated. Neck pain’s precise sources and causes are frequently revealed by medical exams or diagnostic imaging [12].

Physiotherapists use several treatment techniques for the management of CMNP, which include mobilisation, therapeutic exercises [13,14], mechanical traction [15], and electrotherapy [16]. Exercise therapy is one of the most practised therapy techniques for managing CMNP. Stretching, mobilising exercises, resistance training, and proprioceptive and endurance training are just a few techniques used in exercise therapy, many of which require the patient’s active participation [17]. Without regard to the kind of exercise, this approach is effective for non-specific chronic neck discomfort [18]. Two trials that provided moderate-quality evidence suggest that neck pain exercise programs, such as active cervical rotations, weight training, and stretching, may improve pain and neck functions but not instantly the quality of life [13]. Neck pain can be reduced with specific moderate- and high-intensity workouts for cervical and shoulder muscles. Additionally, CMNP patients will benefit from increased neck muscle strength and motion range due to intensive, frequent training, which will reduce disability and improve function [11]. For both short- and long-term subacute and chronic mechanical neck diseases, exercise is moderately helpful when performed alone and effective when combined with mobilisation or manipulation [11,19].

Trigger point dry needling (TrP-DN) is an invasive technique in which solid-filament needles are inserted into the trigger point, characterised as a hyperirritable area or nodule of exquisite tenderness to palpation, known as the myofascial trigger point (MTrP). According to Borg-Stein [20,21], TrP can exist in the upper trapezius and the muscles around neck and shoulder region in neck pain patients. A few researchers reported that MTrPs might be the prominent cause of neck pain in CMNP patients [22]. Dry needling (DN) is a non-pharmacological technique commonly applied for MTrP-related pain relief in the myofascial pain syndrome. It has been reported that DN can cause a relief in pain secondary to MTrPs by producing local twitch responses (LTRs) [23]. Many researchers [24,25,26] have found that the effects of DN generally stem from the mechanical disruption of the muscle fibres and nerve endings caused by the needle prick, and few researchers [27] claimed that an LTR is being evoked during the DN procedure. Additionally, recent research has shown that DN of the upper trapezius alleviates myofascial pain syndromes, particularly neck pain. In the study by Perreault, Dunning [27], TrP-DN resulted in immediate pain relief and improved range of motion with a four-week follow-up. 

Numerous studies revealed that TrP-DN is more beneficial compared to sham needling in alleviating neck pain [28,29]. According to a meta-analysis by Kietrys, Palombaro [30], DN is beneficial for rapid pain alleviation in neck pain patients following four weeks of therapy and follow-up. Abbaszadeh-Amirdehi, Ansari [31] found that motor endplate hyperactivity and sympathetic nervous irritability were reduced after only one session of DN insertion into active MTrPs, thereby highlighting the effectiveness of DN in symptom alleviation and active MTrP deactivation. Although several studies have investigated the beneficial effects of DN or exercise therapy independently for CMNP treatment, it is unknown whether the benefits of TrP-DN combined with exercise exceed those of neck exercise alone. Therefore, this study aims to determine the short-term effects of DN with a standardised exercise program on pain and quality of life in patients with CMNP.

## 2. Materials and Methods

### 2.1. Study Design

This study was a randomised control trail which has been registered with clinical trial registry (ClinicalTrials.gov: NCT05220852).

### 2.2. Participants 

Thirty-one participants (five females, twenty-nine males) diagnosed with CMNP, with a mean age of 35 ± 10 years, body mass of 86 ± 19 kg, and height of 173 ± 8 cm, took part in the study. The sample size was estimated as 30 (15 per group) on the basis a randomised control trial which studied the efficacy of DN injection and taping on pain and quality of life in CMNP [1]. The details of the sample size calculation are as follows: Group 1 mean = 13.4, standard deviation = 4.9; Group 2 mean = 6.9; ratio between the groups = 1:1 (*p* = 0.05); 80% power; and 0.05 type 1 error. The study was conducted at King Fahad Hospital of the university in Khobar, Saudi Arabia. Patients medically diagnosed with CMNP with the presence of MTrP in the neck or shoulder region were included in the study. Those who were confirmed to have cervical radiculopathy or myelopathy, needle phobia, a history of cervical/shoulder/whiplash injuries, ‘red flag’ signs, congenital deformities, any therapies the patient might have taken during the inclusion in the study, and fibromyalgia or any contraindication for DN such as anticoagulant medication or mental disorders were excluded from this study. An independent researcher who was not associated with the current research allocated the participants into experimental and control groups. Figure 1 shows the consort diagram regarding the participant’s flow. All the participants provided their written informed consent, and the research was approved by the Institutional Review Board of Imam Abdulrahman Bin Faisal University (IRB-PGS-2021-03-209).

### 2.3. Evaluation

After being allocated either to the control or experimental group, the baseline measurements were taken prior to the first intervention. The participants were asked to fill out questionnaires about their quality of life, pain intensity, functional neck disability, and depression levels.

The Arabic version of the Numeric Pain Rating Scale (NPRS) is proven to be reliable and valid [32] in assessing pain intensity levels in musculoskeletal disorders. The NPRS ranges from 0 to 10 (0: minimum pain; 10: maximum pain) and is an effective tool for determining a patient’s pain intensity level [33].

In addition, the Arabic version of the Neck Disability Index (NDI), which is highly reliable and valid [34], was used to determine the participants’ neck functional disability levels. The NDI is intended to measure the pain intensity level in the neck and how it impacts the daily activities and quality of life of the patents. The NDI comprises ten questions with a total score range of 0 to 54 [35]. 

Furthermore, the Arabic version of Ware and Sherbourne’s (1992) Short Form-36 Quality of Life Scale (SF-36 QOLS) [36], which has been validated [37], was used to assess the quality of life of the participants. The SF-36 QOLS has eight subsections that assess the body’s physical functions through analyses of “body pain, physical role difficulties, vitality/energy, the general perception of one’s health, social functions, and mental health”. The score for each subsection was calculated separately and ranged from 0 to 100 (0: worst health status; 100: best health status). The study also obtained scores for two primary health components: physical and mental health. 

Moreover, the Arabic version of Beck Depression Inventory (BDI) [38] was used to measure the patients’ depression levels. The BDI total score runs from 0 to 63, with a higher score indicating a more severe depression [39]. The patients were asked to answer the instrument again 10 min following the final intervention (post-intervention) with the same pre-intervention questionnaire.

**Figure 1 jcm-11-06167-f001:**
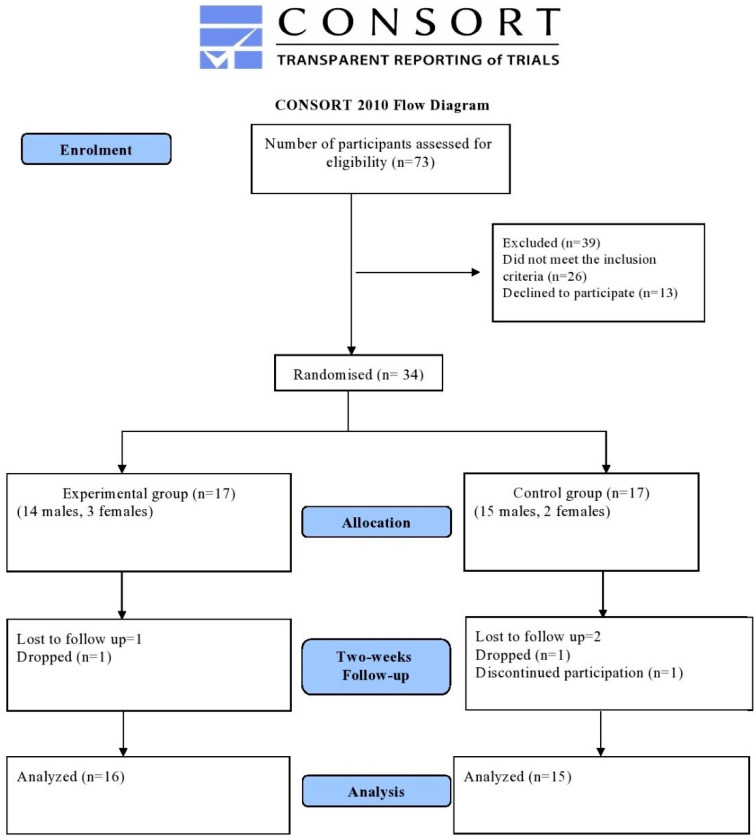
Consort diagram of participant flow.

### 2.4. Procedure

#### 2.4.1. Dry Needling Intervention 

The experimental group received one DN session and underwent an exercise program for two weeks. DN was applied to the overactive TrP of the upper trapezius muscle. The participants were asked to be in a prone lying position, and the needle was inserted into their skin until the first LTR was elicited. The needle was then moved upwards and downwards at around 1 Hz (2–3 mm vertical motions, without rotations) for 25–30 s. Based on healthcare evidence, TrP-DN is the most efficient practice if LTR is elicited in needling therapy [1,40].

The TrP-DN was performed using stainless steel non-reusable needles (0.3 40 mm; Novasan, S.A.). The “fast-in, fast-out needle insertion” method proposed by Hong [41] was used, and a TrP area was palpated and marked for needle insertion into the upper trapezius muscle. The insertion was performed until an LTR was elicited, as the needles were placed into the symptomatic side of the neck at the region of the active TrP and penetrated the skin for 10–15 mm [1]. The patients were followed up for two weeks. From the day after the needling, the participants started the exercise program (Table 1). The needling was performed by a certified DN practitioner.

#### 2.4.2. Neck Exercise Program

The experimental and the control group underwent the standardised neck exercise program consisting of strengthening exercises and stretching the muscles around the neck and upper back region, scapular muscles retraining etc. [1]. The details of the exercise program are presented in Table 1. The primary investigator supervised all the exercise sessions.

**Table 1 jcm-11-06167-t001:** Exercise protocol used in the study for both control group and experimental group.

Exercise	Description
Spinal flexion (Craniocervical region)	- For the neck and upper-back muscles - training to improve holding capacity of neck flexors - re-education of craniocervial movements - 3 sets of 10 repetition
Spinal flexion (Craniocervical region)	- For the neck and upper-back muscles - Neck extensors’ training holding capacity - 3 sets of 10 repetition
Shoulder elevation	- For upper-back and shoulder muscles - 3 sets of 10 repetition
Retraining of scapular muscles	- For the upper-back muscles - Endurance exercise of stabiliser of scapula - retraining scapular orientation in neutral position - Retraining of scapular control with weight and arm motion - 3 sets of 10 repetition
Side bending (right and left) exercise	- 3 sets of 10 repetition
Neck protrusion	- Isometric resistance provided - 3 sets of 10 repetition
Stretching exercise	- For all cervical muscles - 30 sec of holding
All the exercises performed actively by the patient in sitting position	

### 2.5. Statistical Analysis

IBM SPSS 23.0 was used for the statistical analysis. The normality distribution of the data was determined using the Shapiro–Wilk Test. Independent *t*-tests and paired tests were used for between- and within-group comparisons, respectively, of the normally distributed data. Non-parametric alternatives, two-sample Wilcoxon rank-sum test and Wilcoxon signed-rank test, were used for the categorical variables and for the variables that were not normally distributed. The significance level was ≤0.05, with a 95% confidence interval.

## 3. Results

There were no statistically significant differences in demographic characteristics (age, height, weight) and baseline data between the experimental and control groups (*p* > 0.05) (Table 2 and Table 3).

The post-test NDI, NPRS, SF-36 QOLS, and BDI scores of both the control and experimental groups were significantly different from their pre-test scores (*p* ≤ 0.05) (Table 4 and Table 5)

The between-group comparison of the post-test scores using Wilcoxon rank-sum test at a 0.05 significance level showed that there were no significant differences between the two groups’ NDI, NPRS, BDI and SF-36 QOLS scores (*p* ≥ 0.05) (Table 6). There were no significant differences between the pre-and post-test scores in eight health concepts of SF-36 QOLS (Table 6).

## 4. Discussion

This study aimed to compare the efficacy of DN with a standard exercise program with those of the standard exercise program alone in CMNP patients. To our understanding, the current study is the first research comparing DN with exercise in CMNP patients. DN is frequently used for the management of various musculoskeletal conditions, although its effectiveness in managing CMNP compared to exercise alone is not yet known. This study revealed that one session of TrP-DN with six sessions of a standard exercise program had similar effects to six sessions of the standard exercise program alone in terms of relieving pain, improving quality of life, and alleviating depression in CMNP patients immediately following treatment and at a two-week follow-up. These results support the null hypothesis that DN, coupled with a standard exercise program, has a similar immediate effect on the quality of life and pain intensity level as a standard exercise program alone in CMNP patients. However, for ethical reasons, a true control group (one without any intervention) was included in the study; we are not confident that the aforementioned effect is related to the intervention rather than the passage of time.

The current study results agree with those of previous studies that compared the efficacy of DN with that of other interventions for CMNP. A recent randomised controlled trial (RCT) by Gattie, Cleland [42] on the efficacy of DN and manual therapy (MT) in CMNP revealed that MTrP-DN showed no substantial or clinically meaningful advantages in disability reduction or pain alleviation compared to MT and exercise in both short term and long-term follow-ups. Onat, Polat [1] compared the efficacy of DN and kinesio-tape in mechanical neck pain and found that the latter was superior to DN in improving the flexibility of the neck region and reducing the disability of CMNP patients. Llamas-Ramos, Pecos-Martín [16] compared the short-term effects of TrP-DN and TrP-MT and found that two sessions of TrP-DN had similar pain and disability outcomes to two sessions of TrP-MP. De Meulemeester, Castelein [43] found statistically significant pain and disability reductions in short- and long-term follow-ups in both groups, suggesting that DN was no more effective than MP in alleviating neck/shoulder pain. A recent systematic review by Lew, Kim [44] compared the short- and medium-term effects of ischemic compression or MT and DN on upper back pain, neck pain, and shoulder pain in myofascial pain syndrome patients and found that neither intervention appeared to be superior to the other. Meanwhile, the findings of some studies contrast with ours. A RCT on the short-term effects of DN and exercise on chronic neck whiplash injury revealed that DN reduced pain and disability more than sham needling and exercise [40]. The study included a six-week treatment program with three needling sessions, whereas in our study, each participant had only one DN session.

Additionally, a meta-analysis by Kietrys, Palombaro [30] revealed that TrP-DN effectively and immediately reduced the participants’ neck pain after treatment and at a four-week follow-up, as opposed to sham or placebo. A non-randomised control trial with a three-week follow-up revealed that MTrP-DN resulted in significant pain and disability reduction and mood, function, and TrP status improvement in patients diagnosed with CMNP [23]. However, the participants in the study had three DN sessions per week, unlike the participants in our study, who had only one DN session per week.

The pain reduction as a result of the exercise intervention in this study supports the prior research finding that exercise can help individuals with CMNP [45,46,47]. In our study, the between-group comparison revealed an average reduction of 4.4 points on the pain scale in both groups, and based on the a priori defined minimal clinically important difference of 2.00 points on the NPRS, the average pre-exercise score of both groups was 5.5 and the average post-exercise score was 1. This is similar to the average reduction in the study by Letafatkar, Rabiei [47], in which the exercise group had an average baseline score of 6.54 and an average post-intervention score of 4.27. In the study by Llamas-Ramos, Pecos-Martín [16] on the effects of DN on CMNP, the average pre-DN score was 6.2, and the average post-DN score was 1.9. Meanwhile, level II evidence suggests that therapists integrate strengthening exercises into their treatment plans to alleviate neck pain and improve the quality of life [48].

The NDI scores of both groups in this study were reduced by ~8 points on average from the baseline after exercise. For CMNP, a decrease of five points in the NDI score has previously been established to be a clinically relevant difference [35]. Falla, Jull [49] reported an average NDI score reduction of ~4 points after exercise. In that study, the average NDI score at the beginning of the exercise program was ~15.5, suggesting moderate to severe neck pain. Eftekharsadat, Porjafar [50] reported that in their study, the average pre-DN NDI score was 26.93 and the average post-DN NDI score was 10.38, with an average NDI score reduction of 16.55 points. They also reported an average pre-exercise NDI score of 28.90 and an average post-exercise NDI score of 9.30, with an average NDI score reduction of 19.6 points.

Surprisingly, the SF-36 QOLS scores in all eight health concepts improved in both groups. Falla, Lindstrøm [45] reported a mean pre-exercise overall SF-36 QOLS score of 17 and a mean post-exercise overall SF-36 QOLS score of 11.5, with a mean 6.5-point reduction. In the current study, the average overall SF-36 QOLS score for both groups at the baseline was 76.6, and that of the post-intervention was 87.8, with an average improvement of 11.2 points. Eftekharsadat, Porjafar [50] reported an average pre-DN score of 39.71 and an average post-DN score of 79.85, with an average improvement of 40.14 points. The average exercise group baseline score was 35.70, and the average post-exercise score was 81.60, with an average improvement of 45.9 points. Thus, in chronic pain conditions, there is evidence of a significant correlation between pain severity level and quality of life. This does not prove that quality of life will improve when pain is decreased, but it shows that it is possible [7].

Depression was also alleviated by ~3 points on average in both groups in the current study. The depression levels of all the patients were normal at the baseline, with an average BDI score of 4 points, and the average post-intervention BDI score was 1. The normal depression level ranges from 0 to 13 points [39]; in our study, none of the patients suffered from depression at the baseline or post-intervention. Similar results were reported by Nazari, Bobos [51] regarding the effectiveness of exercise: an average pre-exercise BDI score of 5.5 and an average post-exercise BDI score of 1.6. Meanwhile, Yılmaz, Erdal [52] reported an average pre-DN score of 12.5 and an average post-DN score of 7.

This study did not identify MTrP-DN as an effective intervention, not because we obtained evidence that it is not effective, but because there has been inadequate high-quality research on this subject and unavailable data on the minimum number of sessions required for better CMNP outcomes. Moreover, TrP-DN is considered a painful procedure and causes soreness at the needling site, which can remain for up to two days. As there are contrasting findings in relevant studies regarding the effects of DN, further studies are required. Additionally, four patients who received DN intervention obtained greater cervicogenic headache relief. Two previous RCTs [53,54] revealed similar results regarding the effectiveness of MTrP-DN in cervicogenic headache alleviation. Hence, further investigation of this issue is warranted.

Although no significant adverse events occurred in this study, mild adverse effects occurred in two patients from the DN and exercise groups, including slight bleeding and soreness. This study had a few limitations that must be considered in future studies. First, TrP-DN was performed at only one muscle (the upper trapezius); however, CMNP can also refer to pain from trigger points in other muscles (e.g., splenius cervicis, elevator scapulae, cervical multifidus, splenius cervicis, scalenes, and semispinalis cervicis). Second, we collected data only during a short-term follow-up period (four weeks). Third, DN was applied only in the first session because there were no existing data on the minimum quantity of DN sessions required for CMNP. We did not comment on whether a larger number of DN sessions would indicate more changes in outcomes. Fourth, the current study had a lower proportion of female participants than male ones. In addition, there was no true control group (with no intervention applied) in the study due to ethical issues. The exercises were also performed by the patients themselves; thus, there might have been variations in the level of resistance applied, which may have affected the study results. In future trials, it would be useful to include more than one needling session or one muscle with neck pain. None of the participants in our study suffered from depression at the time of the study. Future studies should include CMNP patients with depressive symptoms.

## 5. Conclusions

In this study, one session of TrP-DN with exercise and exercise alone showed the same pain, depression, and quality-of-life outcomes after a two-week follow-up, and no significant between-group differences were observed. We did not identify TrP-DN as an effective method of treatment, not because we obtained evidence that it is not effective, but because there were inadequate high-quality studies on the subject and unavailable data on the minimum number of injections required for better DN outcomes in CMNP patients.

## Figures and Tables

**Table 2 jcm-11-06167-t002:** Descriptive analysis of demographic factors.

	Group				*p*-Value
	Experimental Group		Control Group		
	Mean	Standard Deviation	Mean	Standard Deviation	
Age	35	12	32	10	0.49
weights	86	19	81	19	0.47
Height	173	8	168	8	0.13

**Table 3 jcm-11-06167-t003:** Between-groups comparison of baseline outcome measures.

Outcome Measures	Pre-Score: Median (IQR)		*p*-Value
	Experimental group	Control Group	
NDI	17 (9.6–22)	14 (8–20)	0.39
NPRS	6 (5–7)	5 (4–6)	0.71
BDI	4 (2–6)	2 (1–6)	0.76
General health	77.3 (62.5–80)	80 (70–90)	0.24
physical functioning	88 (78–95)	90 (85–95)	0.44
Role limitations due to physical health	75 (37.5–100)	75 (50–100)	0.88
Role limitations due to emotional problems	88 (17–100)	100 (100–100)	0.05
Energy/fatigue	60 (48–73)	65 (50–75)	0.72
Emotional well-being	68 (62–76)	72 (64–96)	0.26
Social functioning	75 (75–87.5)	87.5 (75–100)	0.38
Pain	67.5 (61.3–80)	67.5 (57.5–80)	0.52
SF36(overall score)	73.1 (61.7–81)	80.1 (67.3–85.9)	0.20

NDI: Neck Disability Index. NPRS: Numeric pain rating scale. BDI: Beck Depression Inventory. SF36: Short Form-36 Quality of Life Scale.

**Table 4 jcm-11-06167-t004:** Within groups comparison of the control group.

Outcome Measures	Control Group		
	Pre Score (IQR)	Post Score (IQR)	*p*-Value
NDI (post)	14 (8–20)	2.00 (2–4.40)	0.001
NPRS (post)	5 (4–6)	1 (1–4)	0.001
BDI (post)	2 (1–6)	1 (1–3)	0.002
General health (POST	80 (70–90)	84.70 (75–95)	0.001
Physical functioning (POST)	90 (85–95)	90 (90–100)	0.084
Role limitations due to physical health (POST)	75 (50–100)	100 (100–100)	0.005
Role limitations due to emotional problems (POST)	100 (100–100)	100 (100–100)	0.102
Energy/fatigue (POST)	65 (50–75)	85 (80–90)	0.001
Emotional well-being (POST)	72 (64–96)	92 (80–96)	0.004
Social functioning(post)	87.5 (75–100)	100.00 (87.50–100)	0.012
PAIN(POST)	67.5 (57.5–80)	90.00 (90–100)	0.001
SF36(overall score)-post	80.1 (67.3–85.9)	91.20 (85.10–95.50)	0.032

NDI: Neck Disability Index. NPRS: Numeric pain rating scale. BDI: Beck Depression Inventory. SF36: Short Form-36 Quality of Life Scale.

**Table 5 jcm-11-06167-t005:** Within-group comparison of the experimental group.

Outcome Measures	Experimental Group		
	Pre Score (IQR)	Post Score (IQR)	*p*-Value
NDI (post)	17 (9.6–22)	5.00 (2–8)	0.002
NPRS (post)	6 (5–7)	1 (1–2)	<0.001
BDI (post)	4 (2–6)	1(0–4)	0.001
General health (POST	77.3 (62.5–80)	82.50 (75–95)	<0.001
Physical functioning (POST)	88 (78–95)	90(90–100)	0.02
Role limitations due to physical health (POST)	75 (37.5–100)	100 (100–100)	0.008
Role limitations due to emotional problems (POST)	88 (17–100)	100 (100–100)	0.011
Energy/fatigue (POST)	60 (48–73)	80 (80–90)	0.001
Emotional well-being (POST)	68 (62–76)	80 (80–96)	0.015
Social functioning(post)	75 (75–87.5)	87.50 (87.50–100)	0.024
PAIN(POST)	67.5 (61.3–80)	90.00 (90–100)	0.001
SF36(overall score)-POST	73.1 (61.7–81)	84.45(85.10–95.50)	0.058

NDI: Neck Disability Index. NPRS: Numeric pain rating scale. BDI: Beck Depression Inventory. SF36: Short Form-36 Quality of Life Scale.

**Table 6 jcm-11-06167-t006:** Between-group comparison of baseline and mean change of post-test outcomes.

Outcome Measures	Change in Score from Baseline: Median (IQR)		
Post-test change	Group 1	Group 2	*p*-value
NDI	−11.6 (−17–−4)	−12 (−16–−6)	0.905
NPRS	−4 (−5–−3)	−4 (−5–−3)	0.855
BDI	−1 (−3–−1)	−1 (−2–−1)	0.851
General health	2.5 (−2–12.5)	5 (0–15)	0.483
Physical functioning	3 (0–5)	5 (0–10)	0.724
Role limitations due to physical health	12.5 (0–25)	25 (0–50)	0.318
Role limitations due to emotional problems	9 (0–25)	0 (0–0)	0.125
Energy/fatigue	15 (8–20)	15 (10–30)	0.309
Emotional well-being	12 (4–18)	7 (4–20)	0.675
Social functioning	0 (0–12.5)	12.5 (0–25)	0.332
Pain	12.5 (10–27.5)	24.5 (12.5–32.5)	0.163
SF36(overall score)	10.4 (6.5–13.9)	11.6 (6.8–23.7)	0.441

NDI: Neck Disability Index. NPRS: Numeric pain rating scale. BDI: Beck Depression Inventory. SF36: Short Form-36 Quality of Life Scale.

## Data Availability

The data that support the findings of this study are available from the corresponding author upon request.

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
