# Peer review of "Short-Term Effects of Dry Needling with a Standard Exercise Program on Pain and Quality of Life in Patients with Chronic Mechanical Neck Pain"

_jcm, 2022, doi:10.3390/jcm11206167_

Round 1

Reviewer 1 Report

First of all, I would like to thank the journal for choosing me as a manuscript reviewer. On the other hand, I would like to congratulate the authors for their work, however there are many nuances to improve.

This study looks at the benefits of adding an DN to therapeutic exercise in subjects with chronic mechanical neck pain.

The introduction is well presented and structured, although the references are somewhat old. I was surprised to find no mention of Blandpied et al. 2017, which makes one of the most important clinical guides on cervical pain.

In my opinion, the last paragraph of the introduction should be improved, because the objective is mentioned but it comes completely second, even with the previous sentence that DN has already been investigated with therapeutic exercise. This fact makes us think of the need to carry out the study again, it must be differentiated more precisely from the studies that are already published.

In my opinion, the methodology of the study is the one that presents the most deficiencies.

The design of the study is not defined, a fact that should be done at the beginning of the methodology.

Subject selection criteria are mentioned. The contraindications of DN are raised, but we do not know if the subjects included in the study have trigger points or not. You must clarify this fact and how your presence is valued.

The clinical trial registration number does not appear, was the study registered on an RCT platform?

Assessment is presented and defined in the study, although many of the outcome measures remain unclear. I think you should separate each of them into paragraphs and make an effort for the reader to understand it better.

On the other hand, it was not defined who recorded the study variables. Was it the same physiotherapist who performed the intervention who performed the assessment? How much experience did the evaluator have at the time of making this record? How much experience did the physical therapist performing the intervention have?

In my opinion, the statistical analysis should be carried out again. It is said that a t-student was performed and the results are presented with the median and IQR. Were all study variables non-normal?

Conclusion, the authors say that in the within group analysis there are no differences. This is true? the results show the opposite. Are the investigators referring to the results between groups?

Author Response

Authors' response to reviewer's comments

We want to thank all the reviewers for providing us with an opportunity to rewrite the manuscript. Their comments and constructive suggestions helped us to improve this manuscript's quality. The reviewer's comments and authors' responses to the comments are given below.

Reviewer comment #1: The introduction is well presented and structured, although the references are somewhat old. I was surprised to find no mention of Blandpied et al. 2017, which makes one of the most important clinical guides on cervical pain.

Authors' response to reviewer's comment #1:

Thank you for the comments.The reference Blandpied et al. 2017 added to the manuscript

Reviewer comment # 2:

 In my opinion, the last paragraph of the introduction should be improved, because the objective is mentioned but it comes completely second, even with the previous sentence that DN has already been investigated with therapeutic exercise. This fact makes us think of the need to carry out the study again, it must be differentiated more precisely from the studies that are already published.

Authors' response to reviewer's comment # 2:

The last paragraph of the introduction is rewritten for clarity

Reviewer comment #3:

In my opinion, the methodology of the study is the one that presents the most deficiencies. The design of the study is not defined, a fact that should be done at the beginning of the methodology.

Authors' response to reviewer's comment # 3:

The design of the study is added to the manuscript under the subheading 2.1. Study design

Reviewer comment # 4:

Subject selection criteria are mentioned. The contraindications of DN are raised, but we do not know if the subjects included in the study have trigger points or not. You must clarify this fact and how your presence is valued.

Authors' response to reviewer's comment #4: One of the criteria to be included in the study was the presence of palpable trigger point

This has been added to the manuscript as

“Patients medically diagnosed with CMNP with the presence of MTrP in the neck or shoulder region were included in the study. The diagnosis of the MTrP was determined with the following criteria – a) hypersensitive spot in a palpable taut band) palpable or visible local twitch on pincer palpation. c) reproduction of referred pain elicited by palpation of the sensitive spot”

Reviewer comment # 5:

The clinical trial registration number does not appear, was the study registered on an RCT platform?

Authors' response to reviewer's comment # 5:

Clinical trial registration was done for this study. The registration number added to the manuscript

Reviewer comment # 6:

Assessment is presented and defined in the study, although many of the outcome measures remain unclear. I think you should separate each of them into paragraphs and make an effort for the reader to understand it better.

Authors' response to reviewer's comment # 6: Corrected as per the suggestion

Reviewer comment #7: On the other hand, it was not defined who recorded the study variables. Was it the same physiotherapist who performed the intervention who performed the assessment? How much experience did the evaluator have at the time of making this record? How much experience did the physical therapist performing the intervention have?

Authors' response to the reviewer's comment # 7: The intervention was performed by a certified dry-needling practitioner who has more than 5 years of clinical experience. This has already been mentioned in the procedure section of the manuscript under the subheading 2.3- dry needling intervention.

All the exercise session was supervised by the primary investigator who is a trained physical therapist. This also already been mentioned in the procedure section of the manuscript. All the assessment scales used in the study were subjective scales.

Reviewer comment # 8: In my opinion, the statistical analysis should be carried out again. It is said that a t-student was performed and the results are presented with the median and IQR. Were all study variables non-normal?

Authors' response to the reviewer's comment # 8: The  t-test was performed only on the demographic data. The rest of the variables in the study were not normally distributed/ categorical variables. That’s why nonparametric test was performed and the results of those variables presented with the median and IQR

Reviewer comment # 9:

Conclusion, the authors say that in the within-group analysis there are no differences. This is true? The results show the opposite. Are the investigators referring to the results between groups?

Authors' response to the reviewer's comment # 9:

Thank you for the comment. It was a typographical error. It was a between-group analysis. Now corrected.

Reviewer 2 Report

Congratulations for the work done. I have some suggestions that I hope to contribute to the improvement of your study.

INTRODUCTION:

There is repetition of information between the 1st and 2nd paragraph. I suggest putting this information together.

“Neck pain is considered a common public health issue in the general population,

with prevalence rates ranging from 16.7% to 75.1%”

“Neck pain is the most prevalent musculoskeletal condition in the general population

after lower back pain.”

The authors 2 times compare neck pain with low back pain and this can take the reader's focus to the real objective of the study. I suggest removing this information that adds nothing.

“Neck pain is the most prevalent musculoskeletal condition in the general population

after lower back pain.”

 “Geneen, Moore [4] and Leadley, Armstrong [6] found that

CMNP patients have a lower quality of life than people with lumbar disorders, and that

lifestyles influenced the prevalence rates of cervical disorders”

METHODS:

The authors used the Beck Depression Inventory as a measurement instrument, but there is nothing in the introduction to justify its use.

How was the distribution of individuals per group made? Prize draw?

The authors repeat 3 times the information that the duration of the protocol was 2 weeks. Just be written once.

“The patients were followed up for two weeks. The day after needling, the participants started exercising once a day for two weeks (Table 1). The needling was performed by a certified DN practitioner.

Neck exercise protocol

Both the experimental and control groups underwent a two-week standardized neck exercise program”

The information “once a day for two weeks” should be at the item 2.2. Neck exercise protocol.

It would be interesting to describe in the exercise table 1 the individual's position in each exercise and whether the exercise was active or passive.

Usually the evaluation data (measurement instruments) 2.3. Evaluation comes before the intervention 2.2. Procedure.

When they assessed pain using the Numeric Pain Rating Scale, was the pain assessed at the time of assessment, in the last 7 days, in the last month? It was good to add this information.

I would also like to know the period of Neck Disability Index - at the time of assessment, in the last 7 days, in the last month?

RESULTS

Table 2: What is this? Would be Weight and Height?

wights

Hight

This paragraph before the table 6 is unnecessary.

“SF-36 QOLS has eight health concepts: physical functioning, role limitations due to physical health, role

limitations due to emotional problems, energy/fatigue, emotional well-being, social functioning,

pain and general health.”

DISCUSSION

If the users did not experience depression, I do not understand the reason for this assessment. It was suggested to remove the data referring to depression, since the individuals did not present and because the relationship between neck pain and depression was also not justified.

“in our study, none of the patients suffered from depression at the baseline or post-intervention”

Author Response

Authors' response to reviewer's comments

We want to thank all the reviewers for providing us with an opportunity to rewrite the manuscript. Their comments and constructive suggestions helped us to improve this manuscript's quality. The reviewer's comments and authors' responses to the comments are given below.

INTRODUCTION:

Reviewer comment # 1: There is repetition of information between the 1st and 2nd paragraph. I suggest putting this information together.

“Neck pain is considered a common public health issue in the general population,

with prevalence rates ranging from 16.7% to 75.1%”

“Neck pain is the most prevalent musculoskeletal condition in the general population

after lower back pain.”

Authors' response to the reviewer's comment # 1:

Corrected as per the suggestion

Reviewer comment # 2: The authors 2 times compare neck pain with low back pain and this can take the reader's focus to the real objective of the study. I suggest removing this information that adds nothing.

“Neck pain is the most prevalent musculoskeletal condition in the general population

after lower back pain.”

 “Geneen, Moore [4] and Leadley, Armstrong [6] found that

CMNP patients have a lower quality of life than people with lumbar disorders, and that

lifestyles influenced the prevalence rates of cervical disorders”

 Authors' response to the reviewer's comment # 2:

 Corrected as per the suggestion

METHODS:

Reviewer comment # 3: The authors used the Beck Depression Inventory as a measurement instrument, but there is nothing in the introduction to justify its use.

 Authors' response to the reviewer's comment # 3: The introduction has been updated for justifying the Beck Depression Inventory as a measurement instrument

Reviewer comment# 4: How was the distribution of individuals per group made? Prize draw?

 Author’s response to the reviewer's comment # 4 : the distribution of individuals per group was made by an independent researcher who is not part of our study by using sealed envelops

Reviewer comment # 5: The authors repeat 3 times the information that the duration of the protocol was 2 weeks. Just be written once.

“The patients were followed up for two weeks. The day after needling, the participants started exercising once a day for two weeks (Table 1). The needling was performed by a certified DN practitioner.

Neck exercise protocol

Both the experimental and control groups underwent a two-week standardized neck exercise program”

The information “once a day for two weeks” should be at the item 2.2. Neck exercise protocol.

 Authors' response to the reviewer's comment# 5 :

 Corrected as per the suggestion

Reviewer comment # 6: It would be interesting to describe in the exercise table 1 the individual's position in each exercise and whether the exercise was active or passive.

 Authors' response to the reviewer's comment # 6 : individual's position in each exercise and whether the exercise was active or passive added in the table as footnote

Reviewer comment # 7: Usually the evaluation data (measurement instruments) 2.3. Evaluation comes before the intervention 2.2. Procedure.

 Authors' response to the reviewer's comment # 7:

 Corrected as per the suggestion

Reviewer comment # 8:

When they assessed pain using the Numeric Pain Rating Scale, was the pain assessed at the time of assessment, in the last 7 days, in the last month? It was good to add this information.

 Authors' response to the reviewer's comment # 8:

The pain was assessed at the time of assessment

Reviewer comment # 9: I would also like to know the period of Neck Disability Index - at the time of assessment, in the last 7 days, in the last month?

 Authors' response to the reviewer's comment # 9:

At the time of the assessment

RESULTS

Reviewer comment # 10: Table 2: What is this? Would be Weight and Height?

wights

Hight

 Authors' response to the reviewer's comment # 10:

It was a typographical error. Corrected

Reviewer comment # 11: This paragraph before the table 6 is unnecessary.

“SF-36 QOLS has eight health concepts: physical functioning, role limitations due to physical health, role

limitations due to emotional problems, energy/fatigue, emotional well-being, social functioning,

pain and general health.”

 Authors' response to the reviewer's comment # 11:

Thank you for the comments. Removed as per the suggestion

DISCUSSION

Reviewer comment # 12: If the users did not experience depression, I do not understand the reason for this assessment. It was suggested to remove the data referring to depression, since the individuals did not present and because the relationship between neck pain and depression was also not justified.

“in our study, none of the patients suffered from depression at the baseline or post-intervention”

Authors' response to the reviewer's comment# 12 : This has been added as a limitation of our study